# A framework to determine active neurons and networks within the mouse brain reveals how brain activity changes over the course of the day

Guanhua Sun[1], Tomoyuki Mano[1,2,3], Shoi Shi[2,4], Alvin Li[1], Koji L Ode[2], Alex Rosi-Andersen[5], Erica Pedron[5], Steven A. Brown[5†], Hiroki R Ueda[2,6,7], Konstantinos Kompotis[1,5]*, Daniel B. Forger[1,8]*

1 Department of Mathematics, University of Michigan, Ann Arbor, Michigan, United States, 2 Department of Systems Pharmacology, Graduate School of Medicine, The University of Tokyo, Tokyo, Japan, 3 Computational Neuroethology Unit, Okinawa Institute of Science and Technology Graduate University (OIST), Okinawa, Japan, 4 International Institute for Integrative Sleep Medicine (IIIS), University of Tsukuba, Tsukuba, Ibaraki, Japan, 5 Chronobiology and Sleep Research Group, Institute of Pharmacology and Toxicology, University of Zurich, Zurich, Switzerland, 6 Laboratory of Synthetic Biology, RIKEN Biosystems Dynamics Research, Osaka, Japan, 7 Department of Systems Biology, Institute of Life Science, Kurume University, Fukuoka, Japan, 8 Gilbert S. Omenn Department of Computational Medicine and Bioinformatics, University of Michigan, Ann Arbor, Michigan, United States

† In memoriam
* forger@umich.edu (DBF); konstantinos.kompotis@pharma.uzh.ch (KK)

## Abstract

The mouse brain's activity changes drastically over a day despite being generated from the same neurons and physical connectivity. To better understand this, we develop an experimental-computational pipeline to determine which neurons and networks are most active at different times of the day. We genetically mark active neurons of freely behaving mice at four times of the day with a c-Fos activity-dependent TRAP2 system. Neurons are imaged and digitized in 3D, and their molecular properties are inferred from the latest brain spatial transcriptomic dataset. We then develop a new computational method to analyze the network formed by the identified active neurons. Applying this pipeline, we observe region and layer-specific activation of neurons in the cortex, especially activation of layer five neurons at the end of the dark (wake) period. We also observe a shift in the balance of excitatory (glutamatergic) neurons versus inhibitory (GABAergic) neurons across the whole brain, especially in the thalamus. Moreover, as the dark (wake) period progresses, the network formed by the active neurons becomes less modular, and the hubs switch from subcortical regions, such as the posterior hypothalamic nucleus, to cortical regions in the default mode network. Taken together, we present a pipeline to understand which neurons and networks may be most activated in the mouse brain during an experimental protocol, and use this pipeline to understand how brain activity changes over the course of a day.

**Data availability statement:** The data used in the analysis for the paper is shared under the following repository: https://figshare.com/s/5c518ccaedc4dd5baeaa. Some raw files are too large to upload and compressed versions are provided. The code used to analyze these data is also uploaded.

**Funding:** We acknowledge the following funding: 1) HFSP (Human Frontier Science Program) RGP0019/2018 to DBF, SB and HU, 2) NSF DMS (National Science Foundation Division of Mathematical Sciences) 2052499 to DBF, and 3) ARO MURI (Multidisciplinary University Research Initiatives) W911NF-22-1-0223 through 5/29/25 to DBF, and 4) Velux Stiftung 1812 to KK. The funders had no role in study design, data collection and analysis, decision to publish, or preparation of the manuscript.

**Competing interests:** I have read the journal's policy and the authors of this manuscript have the following competing interests: H.R.U. is a co-inventor on patent applications covering the CUBIC reagents and a co-founder of CUBICStars Inc.

# 1 Introduction

The physical connectivity of the mouse brain has been mapped with increasing precision, ranging from mesoscopic, regional connectivity [1], to its voxelized extension [2], and further to layer-specific and cell-type-specific connectivity in the thalamocortical system [3]. Most notably, recent efforts have led to the complete reconstruction of a block of the visual cortex, including the wiring diagram of neurons within [4]. While the structural connectivity of the brain and the transcriptomic profiles of its constituent neurons [5,6] are generally considered stable, both the strength of these connections and the activity of individual neurons can vary significantly throughout the day due to sleep and circadian regulation, environmental conditions, and homeostatic needs [7,8]. Consequently, the neural networks governing brain-wide behaviors across different times of the day emerge from the interplay between structural connectivity and the dynamic activation of interconnected neuronal populations.

To understand the hierarchical interaction between different brain regions, researchers have identified key hubs and networks in the mouse brain by studying its structural and functional connectivity [9,10]. For primates in general, three core neurocognitive networks have been suggested to play key roles in the regulation of primate behavior: the salience, default mode, and central executive networks (SN, DMN and CEN) [11]. These networks have more recently been identified in the rodent brain [12–14] and are known to vary in a time-dependent manner [15–18]. While functional connectivity studies have provided valuable insights into the short-term, state-dependent interaction of different brain regions, they fall short in identifying specific neurons or circuits linked to particular behaviors or brain functions. To tackle this limitation, neuronal tagging systems based on genes transcribed shortly after neuronal activation, namely immediate early genes [19], have been developed [20,21]. Some studies have used such tagging systems to reveal regional variation in active neurons [22], but no study to date has provided a basic framework that can study how specific neurons or networks are active at a certain time window during an experimental protocol at the whole-brain level.

To achieve this, we developed a novel experimental and computational framework to identify the most active neurons during different times of the day. We use a Targeted Recombination in Active Populations (TRAP2) system [23] induced by c-Fos expression to label the active neurons in the mouse brain at four different time windows of the day: the beginning (ZT0-4) and the ending (ZT8-12) hours of the resting period of mice, as well as the beginning (ZT12-16) and ending (ZT20-24) hours of the active period of mice. Subsequently, we combine tissue clearing technologies (CUBIC [24]) with light sheet microscopy to attain images with active neurons marked by fluorescent signals. We then applied machine learning techniques to segment individual neurons and align the anatomy to the Allen Brain Atlas Common Coordinate Framework (ABA-CCFv3) [25]. This enables the usage of established spatial transcriptomics data [6] that is used to identify the molecular properties of each active neuron we identified.

Next, we introduced a novel approach that integrates the neuronal information with the mesoscopic structural connectivity data from the Allen Brain Connectivity

Atlas. This approach is based on the assumption that any effective communication between two brain regions requires neurons in both regions to be active and structurally connected. We refer to this derived network as the "active connectivity." We then used established tools from network science [26,27] to analyze the structure of the active connectivity across different time windows, hypothesizing that network hubs, regions with high centrality, would shift over time and play key roles in mediating inter-regional communication. Our analysis revealed notable structural differences in active connectivity across time points, with hub regions at each time window corresponding to brain areas that are known to regulate vigilance and cognitive performance.

## 2 Results

### 2.1 Framework to identify, digitize, and analyze active neurons and networks in the mouse brain

Our framework to identify, digitize, and analyze the active neurons and networks in the mouse brain is divided into three modules:

**Experiment and image acquisition.** To effectively capture and visualize active neurons, we first crossed mice carrying the TRAP2 system (aka Fos2A-iCreERT2; see Methods) with the Ai14 mice (Ai14(RCL-tdT)-D; also see Methods), thereby establishing a line conditionally expressing tdTomato in c-fos expressing cells (Fig 1a, upper panel). More specifically, as spontaneous neuronal activity triggers the expression of the immediate early gene *c-fos*, it allows for the c-Fos-driven expression of a Cre protein which is active only in the presence of hydroxytamoxifen (4-OHT). Meanwhile, the same neuron also expresses the fluorescent marker tdTomato in a Cre-dependent manner, thereby finally tagging neurons that are active simultaneously with 4-OHT activity (Fig 1a, bottom panel). Employing this mouse line to examine how active brain connectivity is modified to underlie behavior throughout the 24-hour day, we designed our experiment to tag neurons that are active specifically at circadian windows with known opposing behaviors, such as rest, arousal, and response to the presence of light (Fig 1b). We thus TRAPped neurons at the beginning (ZT0-4) or close to the end (ZT8-12) of their resting period (light period), as well as at the beginning (ZT12-16) or near the end (ZT20-24) of their active period (dark period). After the experiment, all mouse brains underwent extraction, clearing according to the CUBIC tissue clearing pipeline [24], and image acquisition with mesoSPIM (mesoscale selective plane illumination microscopy; [28]) (Fig 1b, bottom right).

**Neuron segmentation and reconstruction.** After obtaining the fluorescent images (see Methods), we used the machine learning algorithm used in [29] to segment all the *c-fos* expressing neurons based on the fluorescent signals from the stacked 2D images and reconstruct their positions in 3D space (Fig 1c and 1d). Furthermore, we transform the images, as well as the segmented neurons, based on a SyN algorithm (Methods) to the Allen Brain Atlas—Common Coordinate Framework (ABA-CCF; [25]). Aligning the reconstructed neuronal data to ABA-CCFv3 enables us to use other data that is also aligned to the same framework, including anatomical information of the mouse brain, spatial transcriptomics, and connectivity datasets that are essential for our analysis later.

**Cellomics and network analysis.** After reconstructing and aligning our dataset to the Allen Brain Atlas Common Coordinate Framework (ABA-CCF), we developed two complementary computational methods to analyze brain-wide neuronal activity by integrating spatial transcriptomic data and mesoscopic structural connectivity. The first method aligns the spatial distribution of active neurons in our dataset with a reference spatial transcriptomic dataset based on proximity, allowing us to infer the molecular identities of active neurons and examine cell-type-specific activity across different time windows (Fig 1e). The second method constructs an "active connectivity" matrix by integrating regional information of active neurons with the mesoscopic structural connectivity from the Allen Brain Atlas [1]. For each brain, we compute this active connectivity and use established tools from network science to analyze the structural changes of active connectivity across the four periods (Fig 1f). We now present the results derived from our framework.

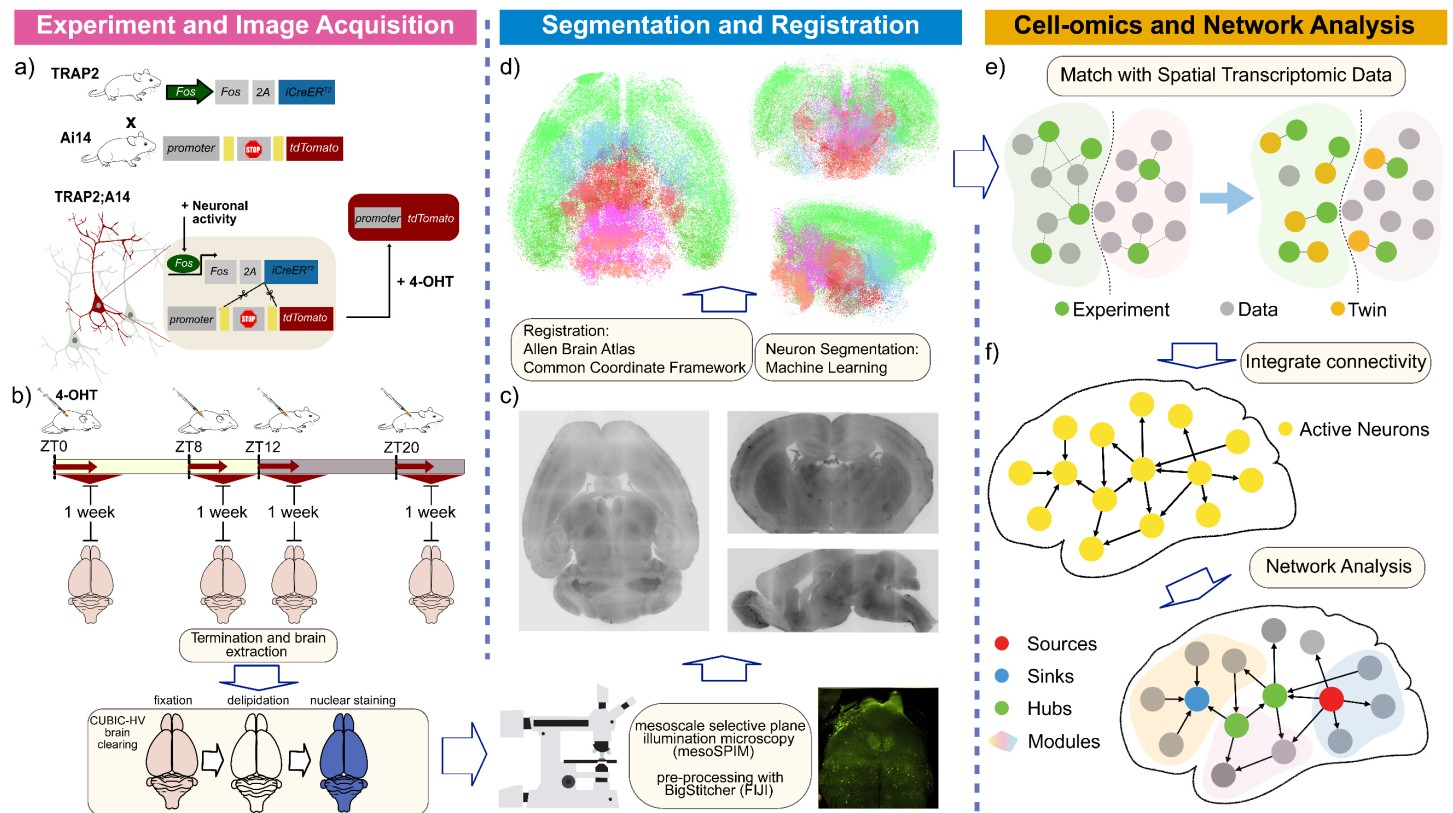

**Fig 1**. **Overall framework to identify, digitize, and analyze the active neurons or networks in the mouse brain.** (a) Schematic of the fosTRAP2 x Ai14 system to capture active neurons. Upper panel: mice expressing the fosTRAP2 system were crossed with mice endogenously expressing Cre-dependent tdTomato (Ai14 line). Bottom panel: Neuronal activity induces expression of fos, which promotes expression of CreERT2. In the presence of 4-OHT, CreERT2 is active, thereby cleaving the stop codon from the tdTomato cassette and enabling its transcription. The result is tdTomato expression only in neurons that were active when the 4-OHT was present. (b) Experimental design for data acquisition at different circadian times. Trapping of neurons was induced by intraperitoneal administration of 4-OHT to fosTRAP2 x Ai14 mice during either the light (ZT0 or 8) or the dark period (ZT12 or 20), as depicted by the red arrows. A week later, mouse brains were harvested and underwent the CUBIC-HV whole brain clearing pipeline, including nucleic staining, prior to image acquisition with mesoSPIM and preprocessing with BigStitcher (FIJI package). (c) Sample fluorescent images of one mouse brain from different view angles (Left: horizontal, right top: coronal, and right bottom: sagittal). d) 3D digital reconstruction of the neurons after segmenting them from the fluorescent images and registering them to the Allen Brain Atlas Common Coordinate Framework. e) Matching neurons in our dataset to the established spatial transcriptomic dataset to identify neuron types. f) Construction and analysis of active connectivity at different time windows.

## 2.2 c-Fos expressing neurons show variation of neuronal activity across different times of the day

After obtaining the 3D fluorescent images, segmenting the neurons and registering them to the ABA-CCF, we obtain the digital 3D reconstruction of neurons of all brains (S1 Fig). With careful observation of all the collected data, we exclude one brain that can not be properly aligned to the Allen Brain Atlas. Moreover, the olfactory area, medulla, and cerebellum of some brains are not intact during the experimental procedure, and therefore excluded from the later analysis.

We pick one brain (ZT0-4a) here as an example to show the spatial distribution of the neurons based on their regional information (S2a-c Figs). Horizontal visualizations of 8 brains, two for each time window, are presented in Fig 2a. We see a variation in the total number of neurons that are identified, with most occuring at ZT20-24 and ZT 0-4 (Fig 2b). To avoid potential discrepancies between different images based on the experimental procedures, we calculated the ratio of the number of neurons in each region to the total number of neurons in a brain to investigate its regional composition (Figs 2c and S2). Among the different major brain regions, we found that both the proportion and absolute number of active

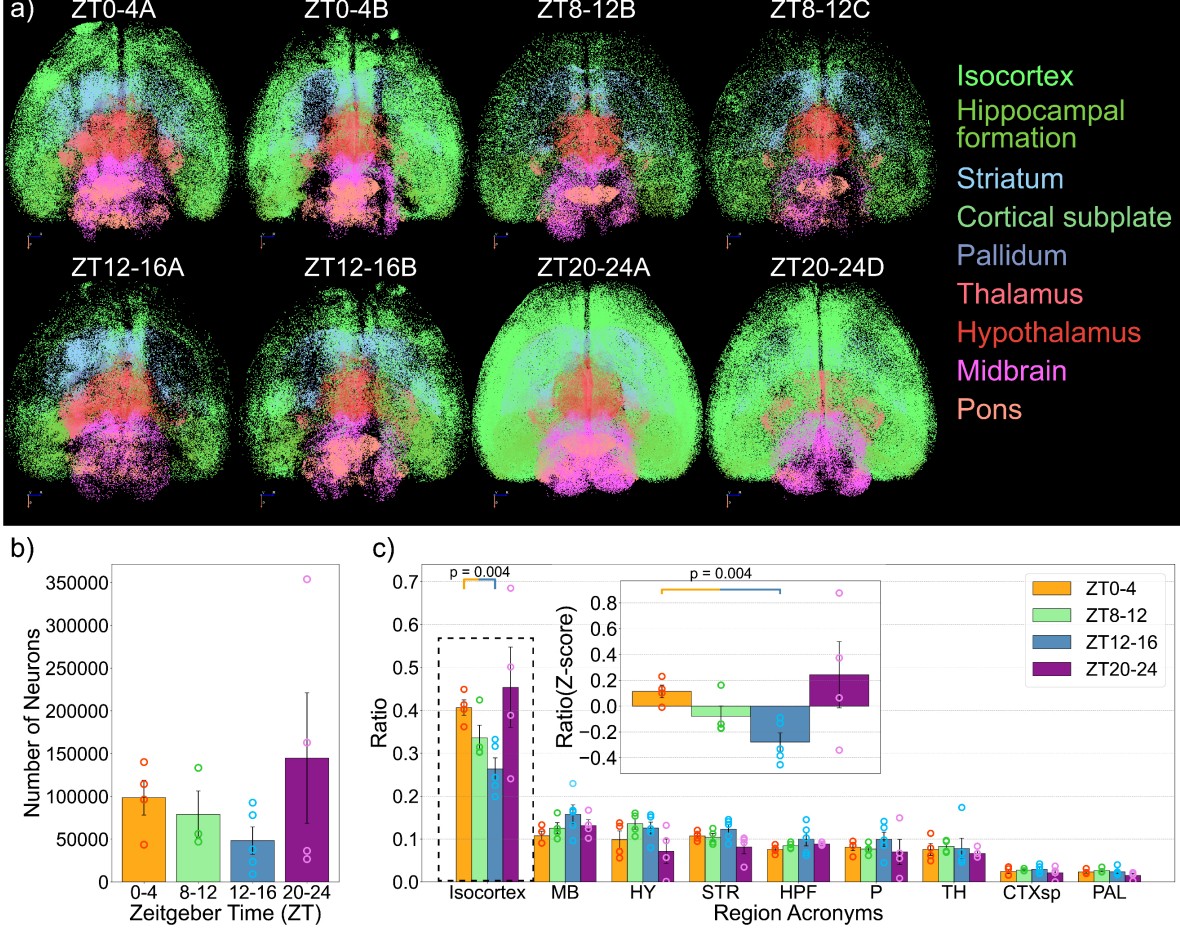

**Fig 2.** **Variation of active neurons in the whole brain across different time windows.** (a) Visualizations of neurons in major brain regions for each time window (2 brains for each time point). The full visualizations of all brains and all brain regions (including the cerebellum, olfactory areas, and the medulla) are presented in S2 Fig. (b) The total number of active neurons in the brain at different time windows during the day. (c) Ratios of different major brain regions for different time windows (ratio is calculated by dividing the number of neurons in each region by the total number of neurons). The ratio of the isocortex is significantly higher at ZT0-4 compared to ZT12-16 (p-value = 0.004), which is further z-scored in the panel inside. The data underlying this figure can be found in S1 Data and at https://figshare.com/s/5c518ccaedc4dd5baeaa.

neurons in the isocortex are significantly higher at ZT0-4, a period with a high amount of sleep compared to ZT12-16, compared to a period when the animals are waking from the rest period (Fig 2c). To our surprise, the isocortex is the only major subregion that shows a significant change in the number of active neurons.

### 2.3 Region and layer-specific distribution of active neurons in the cortex across different times of the day

Since active neurons in the isocortex exhibit significant variation over the four time windows, we continue to examine the distribution of active neurons within the cortex. First, we analyzed the composition of the cortex by calculating the ratio of different cortical subregions to the total cortical population across the four time windows (Fig 3a). This analysis identified several regions with significant activity differences, including the visual area (VIS), the parietal associative area (PTLp), the prelimbic area (PL), and the frontal pole (FRP). We further z-score their ratios to highlight the relative differences in the ratio of active neurons at different time windows (Fig 3b).

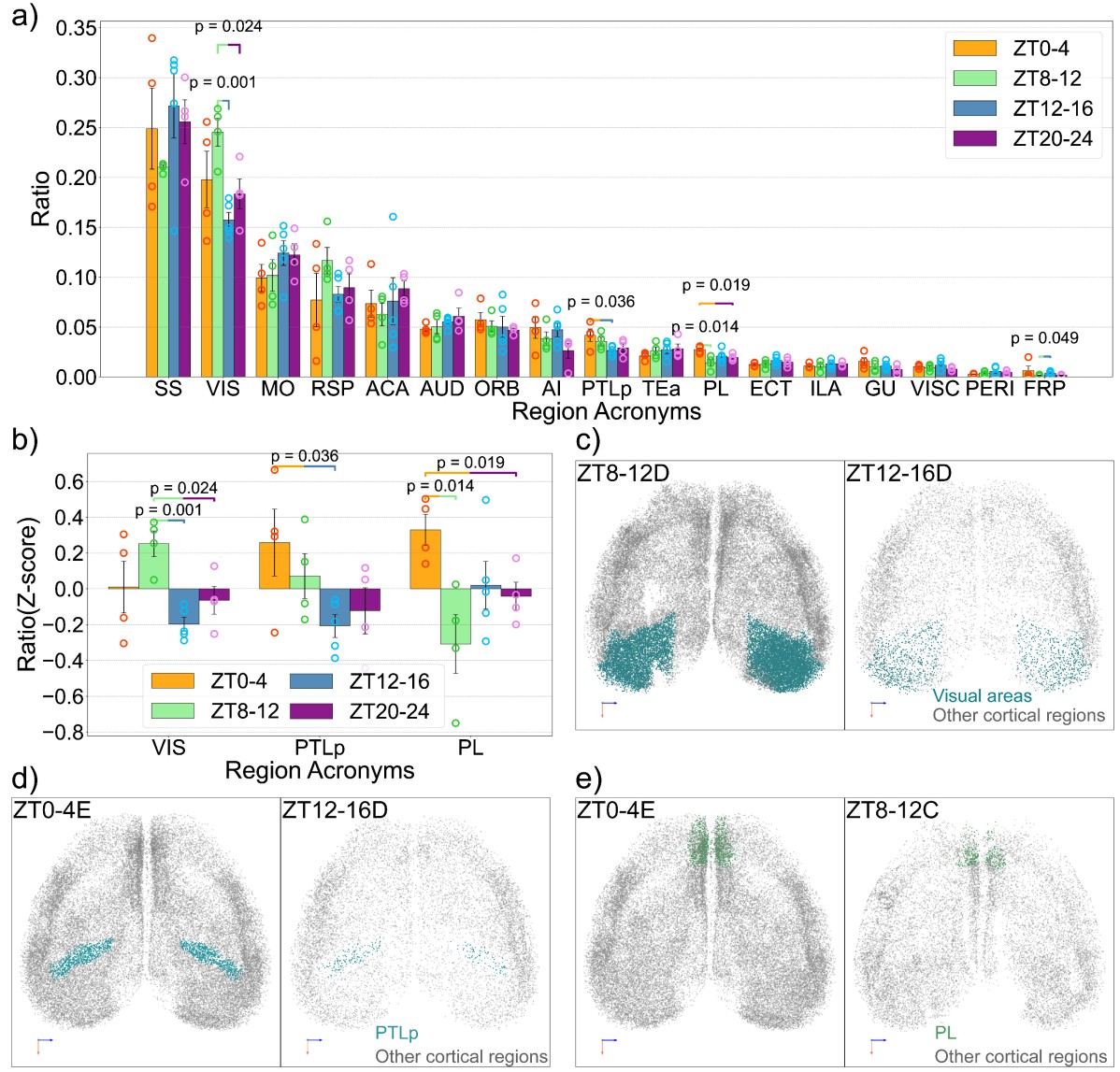

**Fig 3. Region-specific variation of active neuron distribution in the cortex.** (a) Ratios of major cortical regions over the whole cortical population. (b) The three regions that show significant differences in their ratios over the total cortical population across different times (VIS: Visual areas, PTLp: Posterior parietal association area, PL: Prelimbic areas). Here, the ratios are Z-scored based on the average of each region to show the differences between different times. (c) Visualizations of active neurons in the visual regions for ZT8-12D and ZT12-16D brain. d) Visualizations of active neurons in the visual regions for ZT0-4E and ZT12-16D brain. (e) Visualizations of active neurons in the visual regions for ZT0-4E and ZT8-12C brain. The data underlying this figure can be found in S1 Data.

Notably, the visual cortex showed a significant decrease in activity during ZT12-16 compared to ZT8-12, indicating a strong light-off response (Fig 3c). In contrast, the difference between ZT20-24 and ZT0-4 was less pronounced, suggesting a weaker light-on response, a phenomenon previously reported in the literature [30]. On average, visual areas were more active during the light period (ZT0–4 and ZT8–12) than the dark period (ZT12–16 and ZT20–24). The adjacent posterior parietal association area (PTLp) also has higher activity during the light period (Fig 3d), which also has been reported [31]. However, PTLp displayed a more balanced light-on and light-off response. Unlike VIS or PTLp, the

prelimbic area (PL) shows significant differences between ZT0–4 and ZT8–12 (Fig 3e), corresponding to the beginning and the end of the light period, where the mice are at rest most of the time. To our knowledge, this has not been reported in past studies.

With precise spatial information of all neurons, we can explore these regional differences in greater detail by examining the layer-specific distribution of active neurons for each cortical region. We began the analysis with the primary visual cortex (VISp) by calculating the ratio of different layers (Fig 4a). This revealed striking layer-specific variations across the four time windows. The superficial layers (L1-3) and the input layer (L4) have higher ratios at the beginning of the dark period (ZT12-16) compared to the end of the dark period (ZT20-24). In contrast, deeper layers exhibited the opposite trend, with layer 5 neurons showing a significantly higher ratio at the end of the dark period (ZT20-24) compared to ZT8-12 and ZT12-16. This pattern is visualized in Fig 4b, highlighting the prominence of the daily variation of active layer 5 neurons in the primary visual cortex.

Surprisingly, this layer-specific variation was widespread in the cortex. For example, the lateral agranular retrosplenial area (Fig 4c) and the temporal association area (Fig 4d) demonstrate a significantly higher ratio of layer 5 neurons. Further analysis in more cortical subregions confirmed this trend (S4a-d Figs). Most of these regions did not show significant changes when analyzed as a whole. However, our analysis reveals that substantial layer-specific reorganization of active neurons has taken place at different times of the day. To quantify this pattern in the cortex, we calculated the overall ratio of neurons of each cortical layer relative to the total cortical population (Fig 4e). This calculation revealed a significantly higher ratio of layer 5 neurons at ZT20-24 compared to other time points, suggesting a general increase in layer 5 neuron activity. Layer 5 neurons are known to play an important role in mediating cortico-cortical communication [32] and initiating slow-wave activity during sleep [33]. Therefore, our findings here suggest increased intercortical communication and elevated sleep pressure near the end of the active period. Although this trend was broadly observed across the cortex, there are some notable exceptions, including the anterior cingulate cortex (ACA) and the anterior insular cortex (AI)

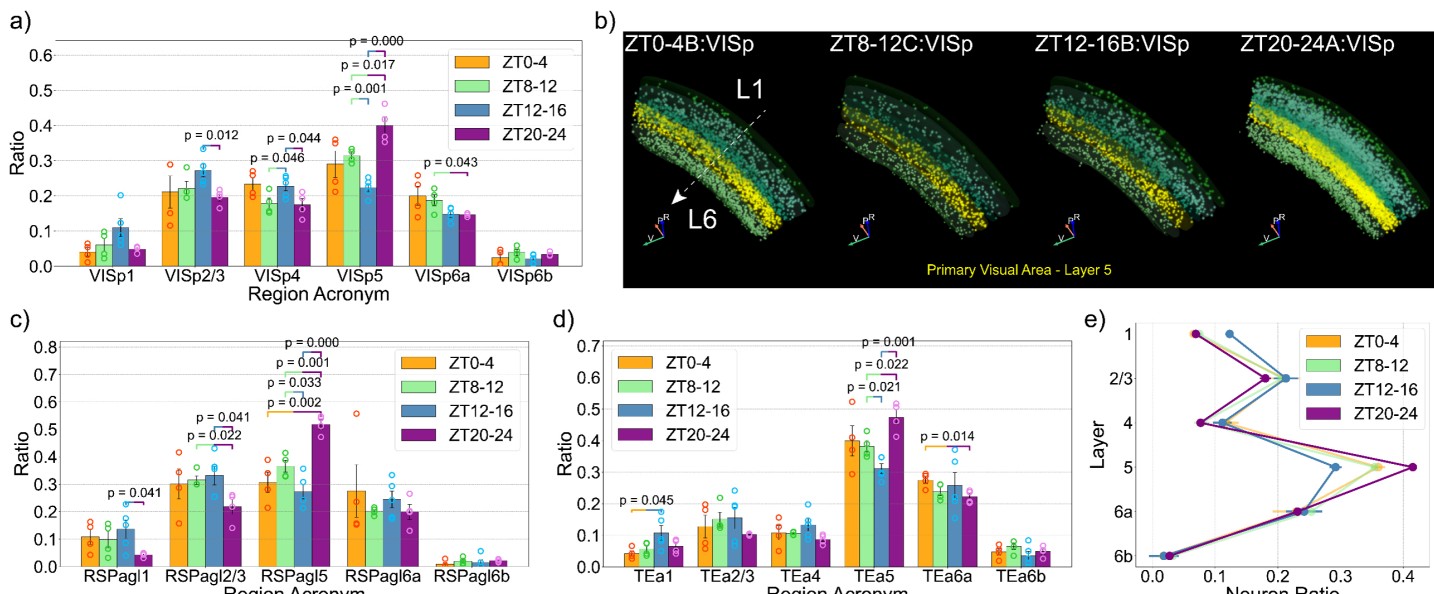

**Fig 4**. **Layer-specific variation of active neuron distribution in the cortex.** (a) The ratio of different layers in the primary visual cortex (VISp). (b) Visualization of active neurons in the primary visual cortex at different times. Layer 5 neurons are emphasized with yellow colors. (c and d) The ratio of different layers in the lateral agranular retrosplenial area (RSPagl) and temporal association areas (TEa). (e) The ratio of neurons in each layer against the total cortical population. The data underlying this figure can be found in S1 Data.

(S4e-h Figs). In particular, both regions(ACA and AI) are core components of the salience network [13], indicating a potential network phenomenon that we will analyze later.

## 2.4 Identifying neuron types with spatial transcriptomic data

Next, we leverage a spatial transcriptomic dataset to identify the properties of the identified active neurons. This dataset(Zhuang-ABA1) provides the exact location with respect to the Allen Brain Atlas CCF, as well as specific molecular information, including the neurotransmitter it releases, of more than 9 million cells across the whole mouse brain [6]. The underlying technique used to generate this dataset, multiplexed error-robust fluorescence in situ hybridization (MERFISH) [34], is not known to be significantly influenced by circadian variation, making it a suitable molecular reference for comparison with our dataset. Because Zhuang-ABA1 has a much higher density of neurons, our dataset can be well immersed with it (Fig 5a). We know the exact location and region of each neuron in both datasets, and thus we can calculate the distance between any neuron in our dataset and the Zhuang-ABA1 data. For each neuron in our dataset, we compute its distance to all neurons within the same anatomical region in the Zhuang-ABA1 data, and assign the closest neighbor within that region as its "twin" (Fig 5b). This approach assumes that neurons of the same type are likely to reside in close spatial proximity. We run this matching algorithm only through the same region, eliminating discrepancies that can happen near the region boundary. As an example, we demonstrate this neuron-matching procedure in the primary visual cortex on the ZT8-12D brain (Fig 5c). To evaluate the quality of these matches, we also examine the distribution of distances

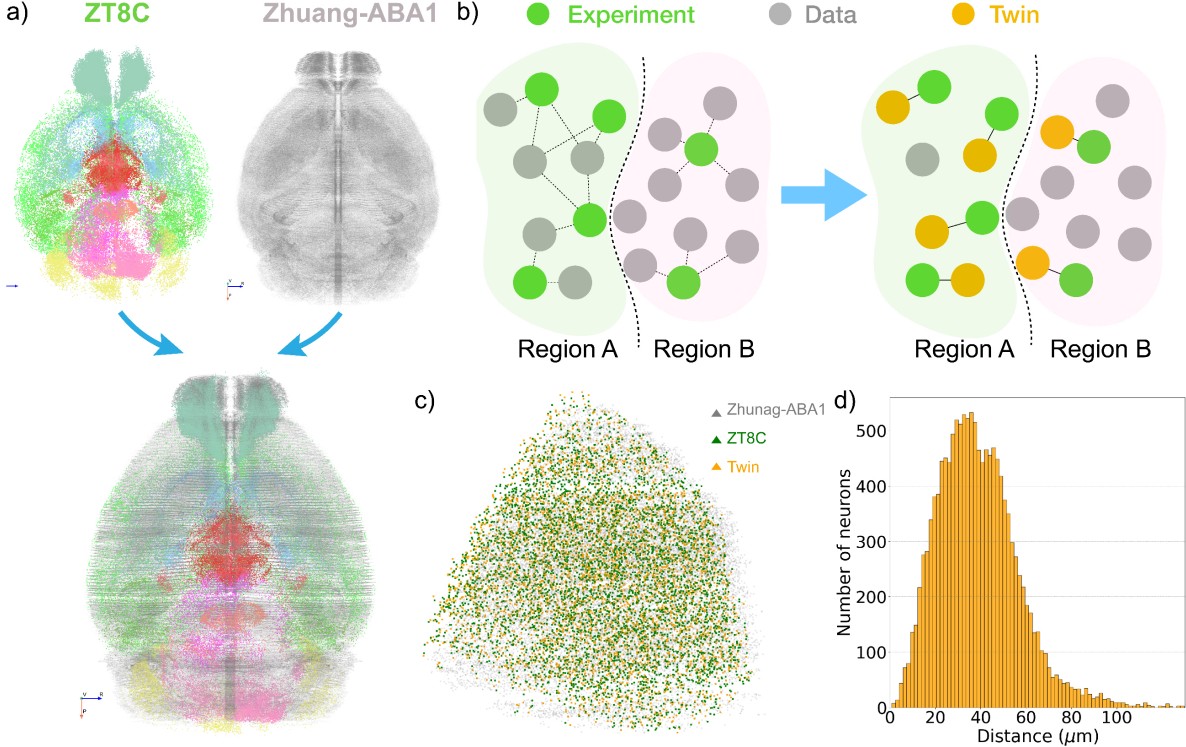

**Fig 5**. **Matching active neurons to the spatial transcriptomic dataset.** (a) Spatial distribution of the neurons in Zhuang-ABA1 (grey), our ZT8C brain (colored based on region) and merged together in the same coordinate framework. (b) Schematics of finding the twin neuron in the spatial transcriptomic dataset (Zhuang-ABA1) for each active neuron in our data based on spatial proximity. (c) A demonstration of the neuron matching process with the right primary visual cortex of the ZT8C brain. Green points are our data, grey points are Zhuang-ABA1 and yellow points are the twin neurons after matching. (d) The distribution of distances between the twin neurons in Zhunag-ABA1 and the neuron in our dataset for the ZT8C brain.

between matched neuron pairs. As shown in Fig 5d, nearly all neurons in our dataset find a match within $100\mu m$, with the majority of matches occurring within $40\mu m$.

After assigning each active neuron in our dataset a "twin" from the reference transcriptomic dataset(Zhuang-ABA1), we assume that the matched neuron shares the same molecular identity as its twin. To validate the accuracy of our neuron-matching approach, we selected two anatomically and molecularly distinct regions: the reticular nucleus (RT), a thalamic region composed almost entirely of GABAergic inhibitory neurons [35], and the ventral tegmental area (VTA), a dopaminergic hub in the hypothalamus [36]. We applied our matching algorithm to the ZT0–4a brain and examined the inferred molecular profiles in these regions (S5a and b Figs). As expected, the method correctly identified a large proportion of GABAergic neurons in the RT, as well as dopaminergic neurons in the VTA (S5c and d Figs). Furthermore, the resulting proportions of GABAergic and dopaminergic neurons in these regions were consistent with values reported in the literature and closely aligned with the distributions observed in the Zhuang-ABA1 dataset, further supporting the validity of our spatial matching framework.

## 2.5 Cell-type-specific variation of active neurons at different times of the day

By applying this method to our dataset, we were able to determine the neurotransmitter profile of all active neurons, enabling a more refined analysis of temporal and regional variation in neurotransmitter composition across the brain. We began by examining the distribution of different neuron types across the entire brain. Notably, we observed a significant shift in the balance between excitatory (glutamatergic) and inhibitory (GABAergic) neurons between ZT12–16 and ZT20–24. The proportion of excitatory neurons was significantly higher during ZT20–24 compared to ZT12–16, whereas the opposite trend was observed for inhibitory neurons (Fig 6a). This shift was also apparent in visualizations in which neurons were color-coded according to neurotransmitter type (Figs 6d and S6).

To further explore the source of this excitatory/inhibitory (E/I) balance shift, we assessed the E/I ratio across major brain regions. Interestingly, the cortical E/I balance remained relatively stable throughout the day (Fig 6b), suggesting that other brain regions—most notably the thalamus—were primarily responsible for the overall change in E/I ratio (Fig 6c). While the cortex as a whole did not show major fluctuations, several cortical subregions did display significant changes in E/I balance, including various regions in the visual cortex (VISp, VISa, VISrl), the ventral anterior cingulate cortex (ACAv), the ventral auditory cortex (AUDv) and the frontal pole (S7 Fig). At the same time, the thalamus exhibited a pronounced reduction in the proportion of active excitatory neurons during ZT20–24, alongside a marked increase in active inhibitory neurons. This regional shift may reflect an increased demand for sleep-related processes late in the day, as inhibitory populations such as those in the reticular nucleus become more active. This result may also be influenced by the relatively uniform neuronal composition observed in regions such as the sensory thalamic nuclei. In contrast, areas like cortical layers 2/3 exhibit a more heterogeneous and intermixed distribution of neuronal subtypes, which may reduce the accuracy of the method.

## 2.6 Construction of active connectivity

Previous observations revealed that active neurons in certain regions, such as the anterior cingulate and anterior insular areas, both components of the salience network, exhibit similar activity patterns. This prompted us to ask whether network-level information could be extracted from our dataset following region-specific analyses of active neurons. To explore this, we leveraged the mesoscopic projectome data from the Allen Brain Atlas, which quantifies the strength of structural connectivity between different brain regions based on viral tracing of EGFP-expressing neurons [1].

To conceptualize communication from one brain region to another, consider the scenario where signals are sent from region A to region B, which has a structural connection density denoted as $\rho_{AB}$. If $\rho_{AB}$ is very low, indicating a lack of structural connectivity, then no signal transmission is expected. However, even when substantial structural connections exist, if region A lacks active neurons, no signal can be initiated. Similarly, if region B lacks active neurons, incoming

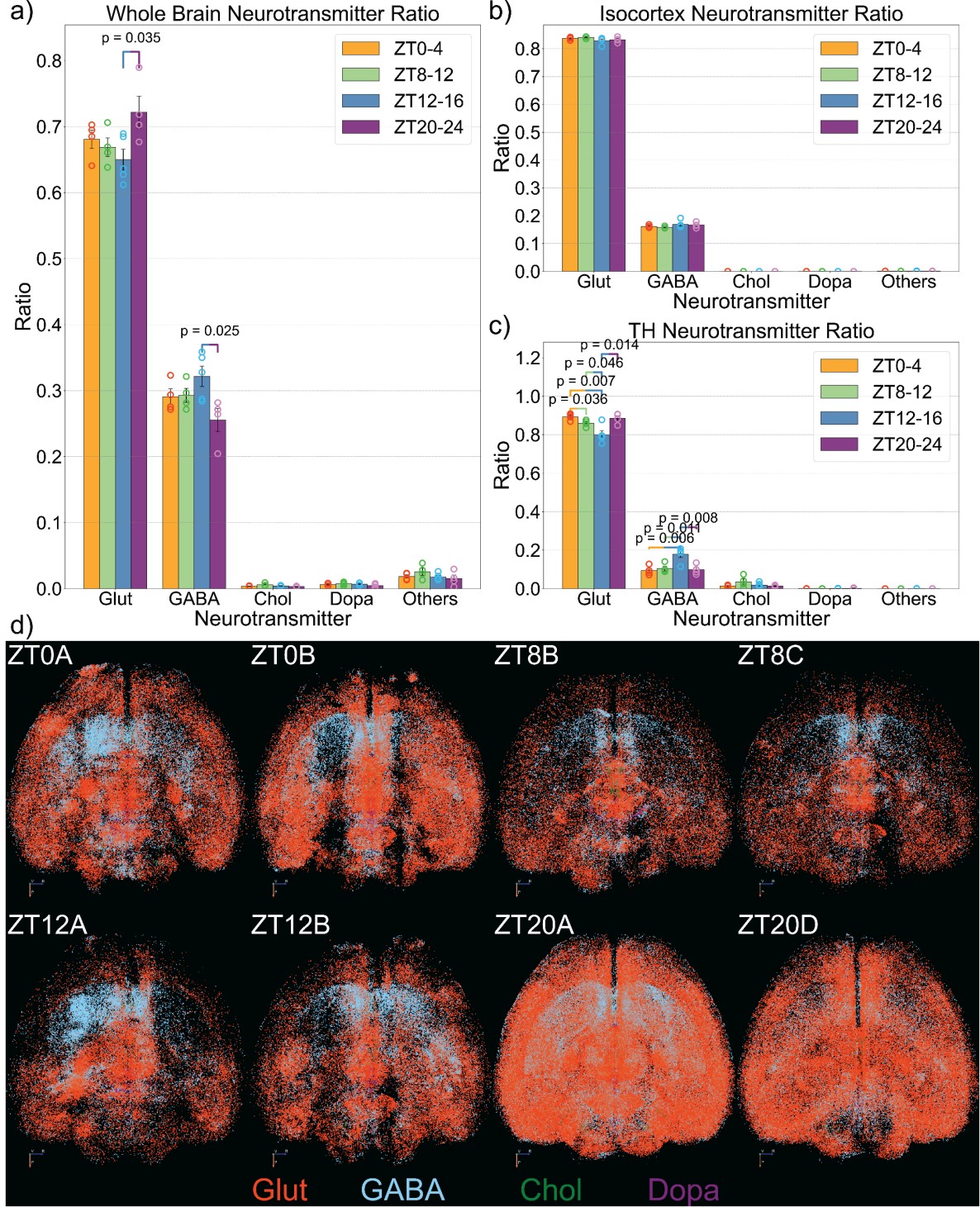

**Fig 6. Cell-type-specific variation of active neurons across the brain.** (a) The ratio of neuronal populations that release different neurotransmitters in the whole brain. (b and c) The ratio of neuronal population of different neurotransmitters in the isocortex and thalamus (TH). (d) Visualizations of neurons based on their neurotransmitter information. We chose two brains for each time segment, which correspond to the brains shown in Fig 2a. The data underlying this figure can be found in S2 Data.

signals may not be propagated further effectively. Based on this reasoning, we define the "active" connection strength from active neurons in region A to those in region B as the product of three factors: the average structural connection density $\rho_{AB}$, the number of active neurons in region A, and the number of active neurons in region B (Fig 7a).

Building on this, we compute a directed, weighted connectivity matrix—termed the active connectivity — by multiplying both the rows and columns of the mesoscopic projectome with a vector representing the number of active neurons of different regions (Fig 7b and Appendix). Each entry in this matrix represents the active connection strength from the active neurons of a source region to those of a target region. Using this method, we computed the active connectivity for each brain at different time windows (Fig 7c) and applied standard network analysis techniques to investigate potential differences in network structure across various time windows (Appendix).

### 2.7 Network changes in active connectivity across different times of the day

Previous studies have analyzed the mesoscopic connectivity [1] using various approaches, offering valuable insights into how the mouse brain is organized. For example, voxel-level analyses have identified sources, sinks, and hubs in the brain [37]; the relationship between physical
connectivity and gene expression has been explored [38]; and the implications of connectivity for wiring cost between regions have also been examined [39]. Here, we extend similar analyses to our active connectivity to investigate the temporal variation of inter-regional communication in the brain across different times of the day. First, we calculate the active connectivity for each brain (S9 Fig) and the average active connectivity matrix at each of the four different time windows. These averages are represented as chord diagrams (Fig 8a-d). We immediately observe that the overall active connection strength between different regions at ZT20-24 are higher compared to the other time points (S10a Fig), a result of more active neurons during ZT20-24. However, this increase in overall connection strength does not dictate the structure of the active network. Here, we use modularity [40], a metric that measures how well a network can be divided into different modules, to characterize the overall structure of the network. By calculating the modularity (See Appendix), we found that the active connectivity during ZT20-24 have significantly lower modularity than the active connectivity for ZT12-16 across different values of $\gamma$, which represents the size of the module we want to identify (Figs 8e and S10). Since ZT12-16 is the beginning of the active period for the mice, and ZT20-24 is the end of the dark period and beginning of sleep. Our findings

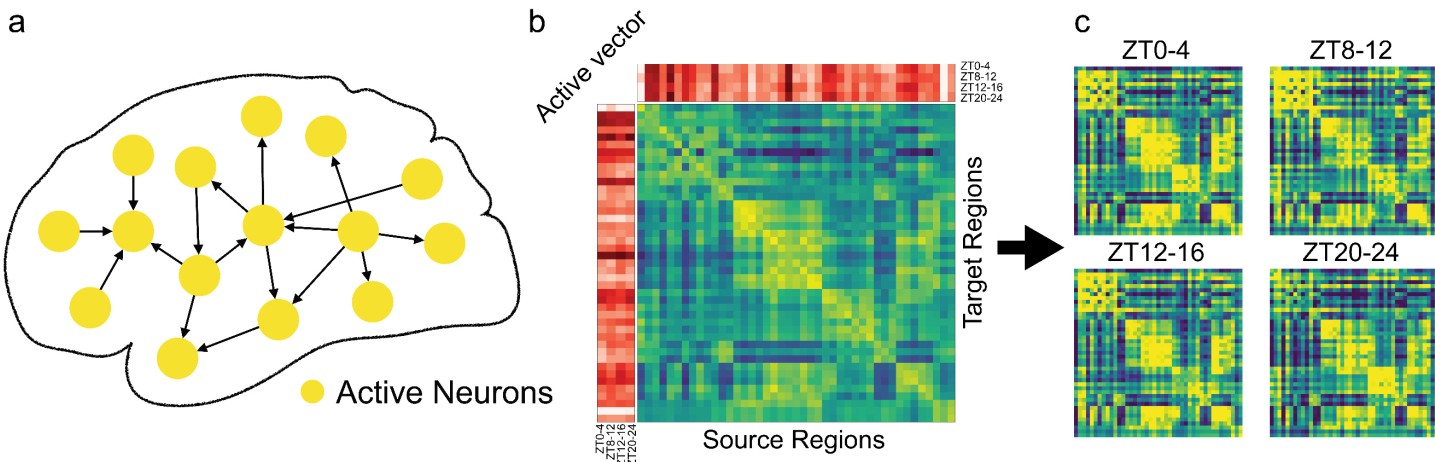

**Fig 7. Construction of the active connectivity.** The active connectivity measures the strength of communication between active neurons in different regions (a). By using the normalized connection density between different regions and the regional information of active neurons (b), we attain different active connectivity matrices for different time windows (c).

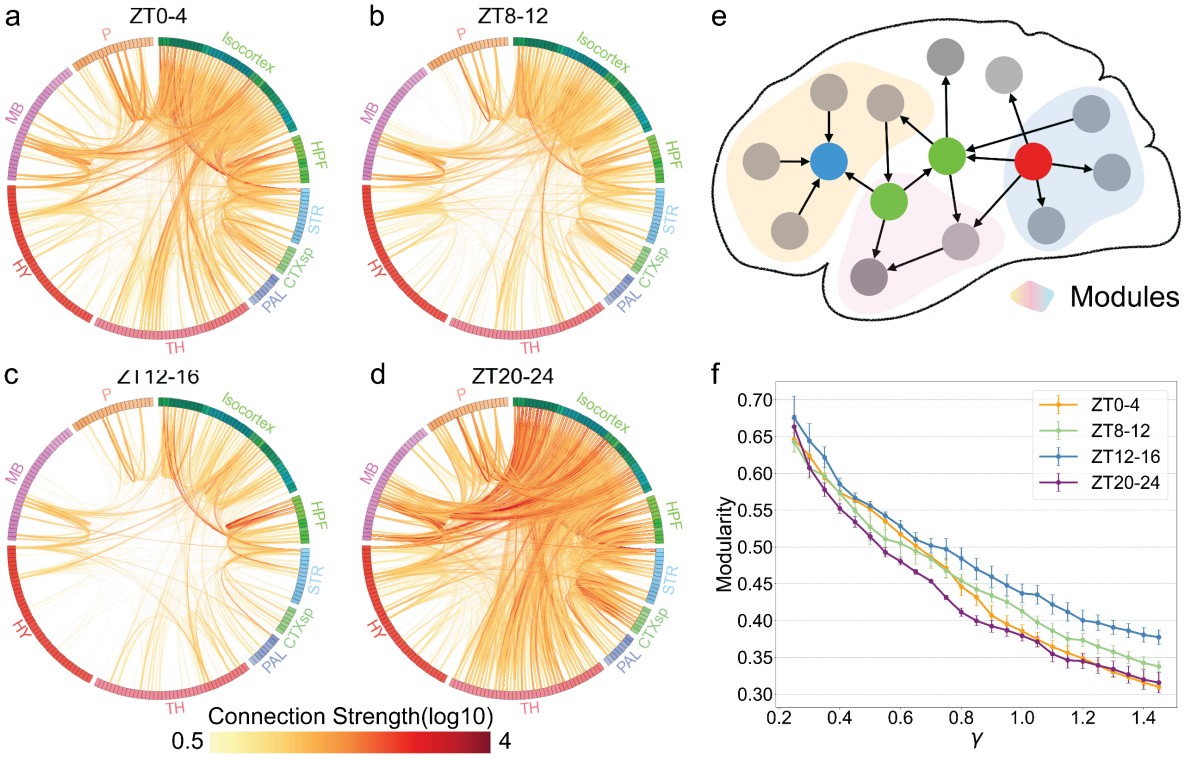

**Fig 8. Structural differences of active connectivity between different time windows.** (a–d) Chord diagrams that visualize the strength of active connections between different regions at different times. (e) Using the active connectivity at different windows, we can identify different modules within the network. (f) Modularity of the whole-brain active for different values of $\gamma$, which represents the size of the modules in the network we wish to identify. The data underlying this figure can be found in S3 Data.

here indicate that the activity throughout the day has strengthened the communication between different brain regions but lowered the overall modularity of the active connectivity.

## 2.8 Spatial variation of hubs in the active connectivity across the day

After analyzing the overall structure of active connectivity, we focused on identifying key regions or networks that play crucial roles at different time points, specifically aiming to detect hubs within the active connectivity (Fig 9a). To identify these hubs, we adopt another network measure: the betweenness centrality, which measures the percentage of each region's involvement in shortest paths across the whole network. A region with high betweenness centrality is understood as a bridge between different parts of the brain, facilitating interaction between otherwise weakly connected modules [26,41,42]. This calculation yields a series of hubs for each time window (S11 Fig). We then average each region's centrality across the brains collected at a time window and highlight the top 10 hubs in the whole brain (Fig 9b); a more comprehensive ranking is provided in S11 Fig. These hubs include both cortical structures, such as the anterior cingulate area (ACA), primary visual cortex (VISp), and retrosplenial cortex (RSP), as well as subcortical regions, including the posterior hypothalamic nucleus (PH) and the motor-related superior colliculus (SCm), which are known to play important roles in movement control, emotion, arousal and other important functions [43,44], but have been ignored in previous network analysis [37].

To further investigate how the spatial distribution of hubs can change, we visualized different regions based on their centrality to illustrate how the spatial distribution of hubs changes over time, as shown in Fig 9c–9f. The most notable shift

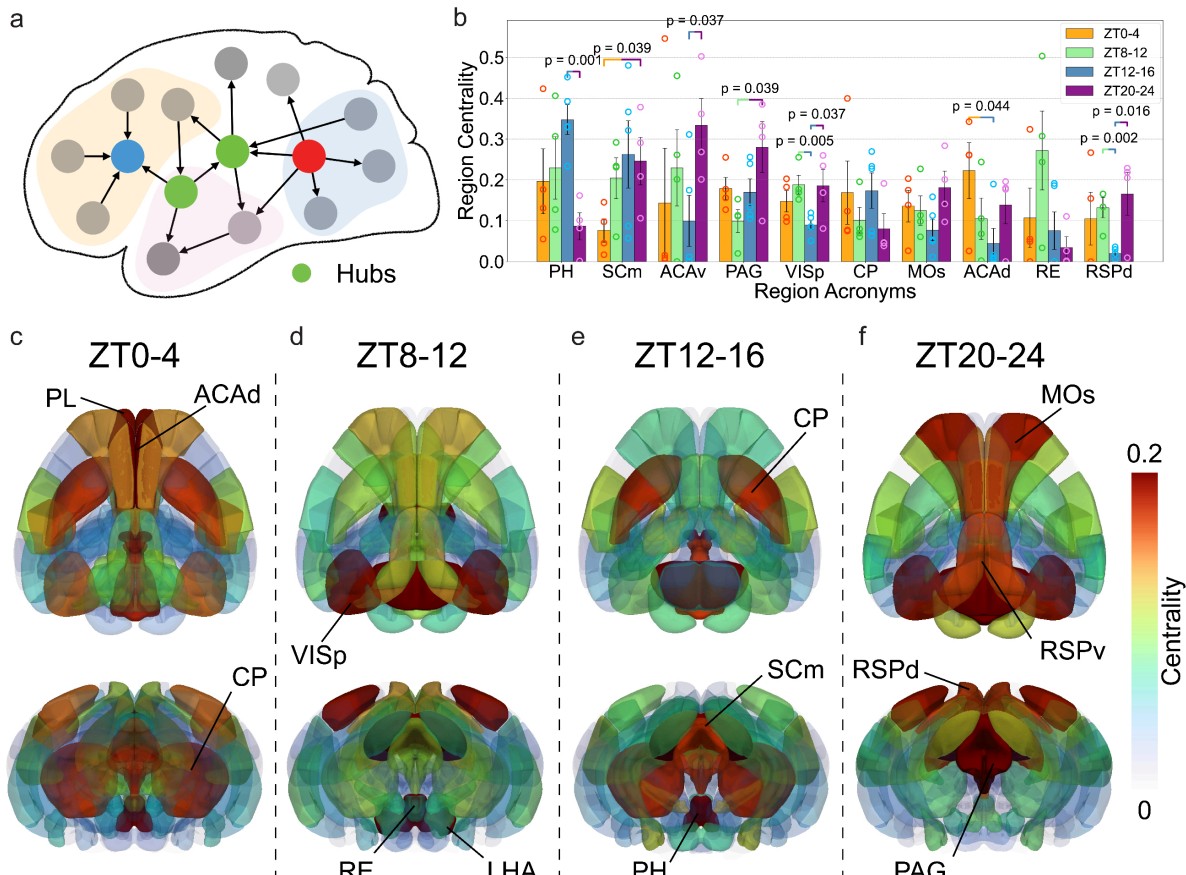

**Fig 9. Active hubs in the brain at different time windows.** (a) We identify the hubs within the active connectivity by calculating the betweenness centrality of different regions. (b) The top 10 regions with the highest centrality on average across the four time windows. (c–f) Visualization of different brain regions, which are colored based on their centrality at different time windows (left to right: ZT0-4, ZT8-12, ZT12-16, ZT20-24) and from different views (top: horizontal, bottom: coronal).

occurs between ZT12–16 and ZT20–24. At ZT20–24, 9 out of the top 15 hubs are cortical regions, whereas only 5 cortical regions appear among the top hubs at ZT12–16. Notably, PH ranks as the most central hub during ZT12–16 but drops to 14th place at ZT20–24. This shift is also evident in Fig 9c: red-highlighted regions (indicating hubs) are primarily located in cortical areas during ZT20–24, while at ZT12–16, the hubs are predominantly distributed within subcortical regions. Moreover, a more detailed examination of the top 15 hubs across time windows revealed a surprising pattern involving the default mode network (DMN) (Fig 10a). In three of the four time windows—ZT0–4, ZT4–8, and ZT20–24—five of the top 15 hubs are DMN regions, whereas at ZT12–16, only two DMN regions appear among the top hubs (S11 Fig). Here we follow the anatomical definition of DMN based on the core regions outlined in [14]. Our findings here indicate that a more central position for the default mode network in the end of the dark period, right before the mice go to rest. This is confirmed by the calculation of the average centrality of all regions in the DMN (Fig 10b), where a significantly higher average is shown at ZT20-24, compared to the value at ZT12-16. Different DMN regions' centrality is also shown in Fig 10c.

It may seem intuitive that an increase in the number of active neurons would correspond to a simultaneous increase in both the strength and centrality of a brain region. However, this is not necessarily the case. For instance, the posterior hypothalamic nucleus (PH) exhibits no significant change in the number of active neurons across the four time windows (Fig 10e). Yet, its regional centrality shifts significantly between ZT12–16 and ZT16–20 (Fig 10f). These findings

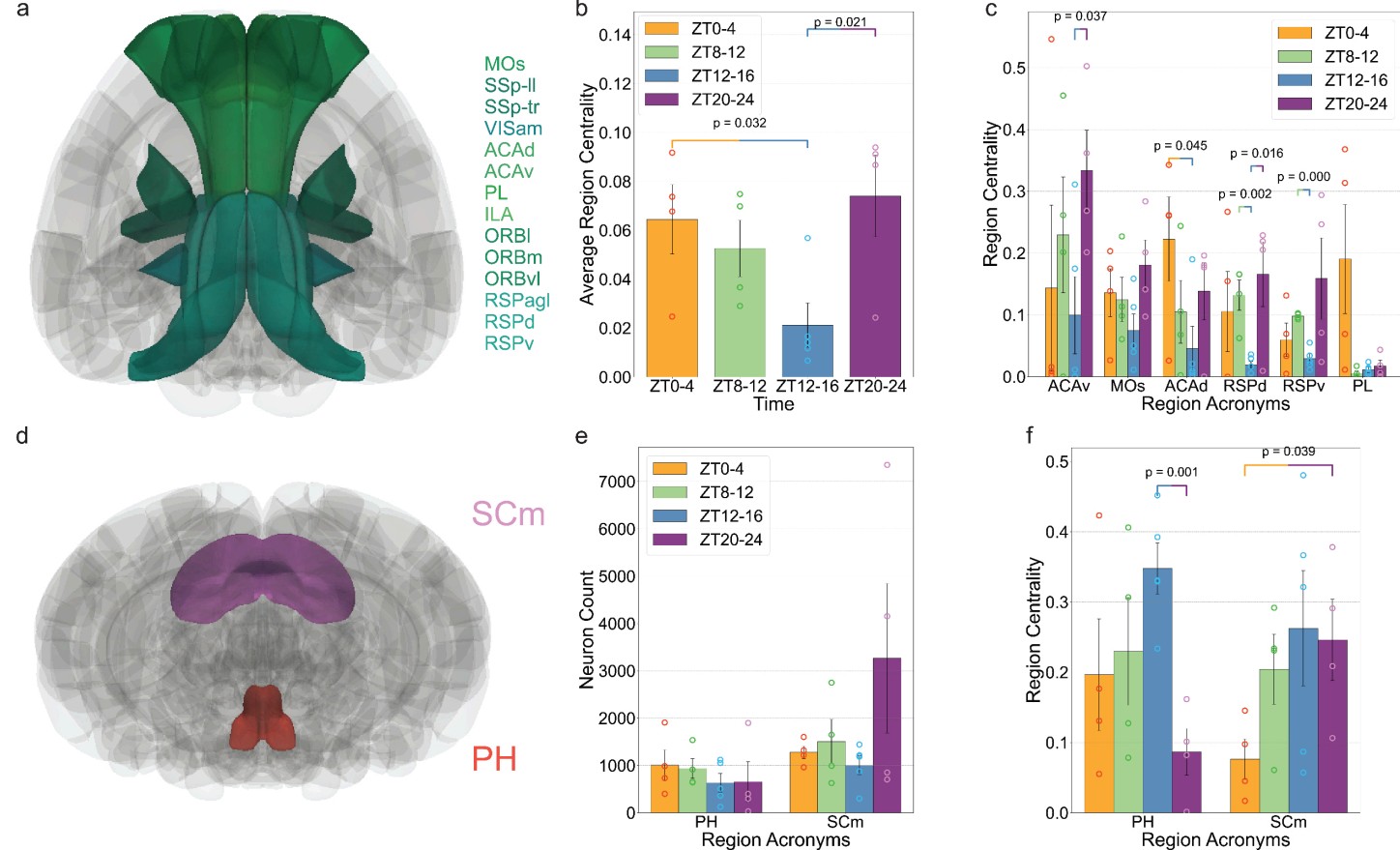

**Fig 10**. **Centrality of the default mode network (DMN), motor-related superior colliculus (SCm) and posterior hypothalamic nucleus (PH).** (a) Positions of different DMN regions in the whole brain (Horizontal view). (b) The average centrality of DMN regions at different time windows. (c) Centrality of the top 6 DMN regions at different time windows. (d) Positions of SCm and PH in the whole brain (Coronal view). (e) Number of active neurons in PH and SCm at different time windows. (f) Centrality of SCm and PH at different time windows.

collectively emphasize the utility of active connectivity, beyond analyzing regions alone, in capturing temporal shifts in brain network organization and identifying key regions whose activities fluctuate through different times of the day.

## 3 Discussion

Here, we introduced an experimental-computational pipeline to study the active neurons and networks in the mouse brain. As a proof of principle, we used the framework to investigate the active neurons and networks in the mouse brain at four different time windows of the day with distinct behaviors. We provided a whole-brain map of active neurons at those time windows, identified layer and region-specific variation of active neurons in the cortex, and observed an excitatory/inhibitory shift of active neurons in the whole-brain. Moreover, our construction of active connectivity has revealed the structural change of inter-regional communication between active neurons across the four time points of the day.

In the current study, we chose to utilize the well-known immediate early gene c-Fos as a driver to endogenously label active neurons. Although other cell types, such as glial cells, can also express c-Fos in the brain [45], previous validation experiments using the original fosTRAP1 system [46] demonstrated that more than 96% of TRAPed cells colocalized with NeuN, a neuron-specific marker. The more evolved fosTRAP2 system employed in our study is equally specific to

neuronal populations, additionally preserving physiological c-Fos function via a 2A-linked iCre fusion [47]. We thus assume that the large majority of captured cells in our study are neurons, and as previously shown, sufficient to recapitulate mammalian behaviors [23]. Based on the number of neurons we identified, our experiments are capturing only about the most active 1% of the total 80 million neurons contained in the mouse brain [48], ignoring the vast majority of inactive neurons. Nevertheless, one of the main strengths of this framework is the modular manner in which it was built, such that it can integrate whole-brain data from both endogenous and exogenous fluorescence, as well as different whole-brain clearing techniques, and match them with novel spatial transcriptomic datasets of interest. To include specific neuronal populations, alternative experimental designs involving labeling of active or inactive populations via staining of expressed proteins of interest can be of complementary value to the analysis conducted here. Moreover, although we chose to focus on four time windows around the 24-hour day with distinct circadian, light and sleep-wake driven behaviors, this framework can also be extended to study mouse brain connectivity under many other conditions, including neurodegenerative states, behavioral interventions, sleep deprivation, or even in combination with automated systems, such as the IntelliCage system [49].

When analyzing our dataset, we observe substantial individual variability in the absolute number of active neurons identified at the same time point. To account for this variability, most comparisons throughout this study are based on the relative ratio of active neurons across different regions—that is, the distribution of active neurons normalized to the total number of identified neurons. We find that these relative ratios, such as the distribution of neurons across cortical layers, are highly consistent among brains sampled within the same temporal window and yield significant and robust results regarding the layer-specific organization of active neurons across time. The layered structure of the cortex is known to have important dynamic consequences [50–52]due to its laminar-specific connectivity patterns [53] and the distinct neuronal types populating different layers [5]. Our findings on the layer-specific distribution of active neurons contribute to this growing body of evidence, highlighting the dynamic modulation of cortical layers across the day.

The regional analysis of active neurons was followed by alignment to the spatial transcriptomic data by finding the closest neighbor in the dataset (Zhuang-ABA1). Although the spatial distribution between neurons in our dataset and the matched neurons is relatively close, the method can still produce discrepancies, in particular, for those regions where different types of neurons are tightly mixed, such as the layer 2/3 of the cortex. In our cell-type-specific analysis of the cortex, we observed that the excitatory-to-inhibitory (E/I) ratio of active neurons remains relatively constant. This finding stands in contrast to several previous studies that report dynamic changes in E/I balance in the cortex [54–56]. This discrepancy can be explained by the error in our method when applied to heterogeneous regions, but can also arise from methodological differences: whereas prior studies typically infer E/I balance from firing activity based on electrophysiological recording, our analysis directly quantifies the number of excitatory and inhibitory neurons. On the other hand, our method will work especially well for brain regions where different types of neurons are more spatially segregated, such as the thalamus or hypothalamus. In the future, we would expect more spatial transcriptomic data with higher resolution that can be seamlessly used in the framework presented in this work to further increase the accuracy of identifying cell properties [57].

We define a new concept, active connectivity, which quantifies the magnitude of inter-regional communication based on underlying structural connectivity and the active neurons in different regions. To construct the active connectivity, we utilized one of the early versions of the whole-brain mesoscopic connectivity dataset [1]. Although a more recent update to this dataset is available [3], providing layer-specific and cell-type-specific projection information, it is limited to the thalamocortical network and therefore was not used for the whole-brain analyses conducted in this study [2,58]. We also did not incorporate the voxelized extension of the connectivity data, which could significantly improve the spatial resolution of the results presented in this work. Integrating these resources represents an important direction for future work. Also, due to experimental procedures, we excluded the olfactory areas, cerebellum and medulla from the analysis. Missing these regions can cause discrepancies in analyzing the brain networks, and thus, including these regions in future studies will be valuable.

Since functional connectivity is a widely used measure for studying interactions between brain regions, here we compare active connectivity with functional connectivity, which are both related to structural connectivity. Compared to the unclear, nonlinear relationship between functional and structural connectivity [59,60], the relationship between active connectivity and structural connectivity is much more straightforward mathematically. Functional connectivity is typically calculated as the correlation between regional activity time series recorded with techniques such as functional MRI (fMRI), resulting in a symmetric (undirected) correlation matrix. In contrast, active connectivity, as defined here, is inherently directed, following the source-target relationships present in the structural connectivity. Furthermore, functional connectivity captures dynamic interactions over short time windows, on the order of seconds to minutes, whereas active connectivity, defined in this work, reflects an integration of neuronal activity over a much longer time frame (approximately four hours in our experiments). Thus, while functional connectivity excels at detecting fast, transient interactions, active connectivity assesses more sustained patterns of network engagement, more causally anchored to the structural connectivity. Nevertheless, it would be valuable for future studies to perform fMRI studies across different circadian time windows and to quantify the corresponding functional connectivity. Comparing these with the active connectivity defined in the present study may yield important insights into the temporal organization of brain networks. From the perspective of neuronal plasticity, mechanisms such as spike-timing-dependent plasticity (STDP) require coincident high-level activity in both pre- and postsynaptic neurons [61]. Since our approach captures the most active neurons in the mouse brain at a given time, the active connectivity defined here may also highlight connections that are most likely undergoing synaptic changes, offering a complementary view of the brain's adaptive network dynamics.

To conclude, multimodal data are available to the field of neuroscience at a surprising speed, creating new avenues for discoveries [62]. We believe that the experimental-computational pipeline presented in this study provides a strong example of how integrating multimodal data can advance the exploration and analysis of the mouse brain.

## 4 Methods

### 4.1 TRAP2 x tdTomato mice

For this experiment, TRAP2 (aka Fos2A-iCreERT2; JAX #030323) mice were bred with Ai14 (aka Ai14(RCL-tdT)-D) mice. Male mice, aged 10–14 weeks and heterozygous for both alleles, were used for experiments. All mice were housed together with their littermates and kept under 12:12 light-dark (LD) cycles with ad libitum access to food and water. Genotyping for the Fos2A-iCreER alleles was performed according to published protocols [23].

All animals were treated according to the regulations and guidelines approved by the Veterinary Office of the Canton of Zurich (licenses ZH141/2020 and ZH068/2023).

### 4.2 4-OHT preparation and administration

For preparation of 4-Hydroxytamoxifen (4-OHT), we followed the Deisseroth lab protocol [63]. Briefly, 4-OHT was dissolved in DMSO and mixed with 25% Tween80 in saline. The final solution was injected intraperitoneally at 20 mg/kg, 250 $\mu$L per animal, within one hour of preparation.

### 4.3 Experimental design

Tagging of active neurons at the four different circadian windows was achieved via i.p. injection with 20 mg/kg 4-OHT in saline at the onset of the respective periods (ZT0, 8, 12, or 20; Fig 1B). After allowing seven days [23] for the expression of tdTomato in captured neurons, mice were terminated with the pentobarbital administration and perfused. Brains were extracted, preserving all structures, and underwent the CUBIC tissue-clearing process.

## 4.4 CUBIC brain clearing

The CUBIC whole brain clearing pipeline was followed as previously described [24]. Briefly, whole brains were immersed in 4% paraformaldehyde (PFA) for 24h, then delipidated and decolored for 5 days in CUBIC-L (CUBIC-HVTM1, TCI chemicals). Subsequently, their cellular nuclei were stained with DAPI for another 5 days, and the refracting index (RI; 1.520) of the transparent brain was calibrated in CUBIC-R+ solution (CUBIC-HVTM1, TCI chemicals). Following the end of incubation and in-between washes, the cleared brains were stored for two days in mineral oil (Mounting Solution (RI 1.520), TCI chemicals) and imaged in the following days.

## 4.5 Whole-brain imaging

The mesoscale selective plane illumination microscopy instrument (mesoSPIM) [64] was used for axially-scanned light-sheet imaging of the cleared brain. Transparent whole-brain images were recorded at 3.2x magnification with an Olympus MVX-10 macroscope with a MVPLAPO objective combined with a Hamamatsu ORCA-FLASH 4.0 V3 camera, at a voxel size of $2.03 \times 2.03 \times 5 \mu m^3$. The laser/filter combinations for imaging were as follows: for registration of autofluorescence and light-scattering, a 405nm (100mW) excitation laser at 100% and no emission filter. For the tdTomato signal, a 561nm (100mW) excitation laser at 50% and a quadruple bandpass (BP444/27; BP523/22; BP594/20; BP704/46) emission filter. Brains were positioned such that the ventral portion was the first plane acquired. Briefly, each whole brain was imaged in 16 tiles per channel (eight tiles per illumination; right or left). The acquired data occupied 400GB per brain on average. Subsequently, images were stitched and fused separately for each channel using the FIJI plugin BigStitcher [65], according to the "MesoSPIM stitching using BigStitcher" guidelines (https://zmb.dozuki.com/Guide/MesoSPIM+Stitching+using+BigStitcher/283). Finally, whole brain images were stored in .tiff format for further processing in the CUBIC pipeline.

## 4.6 Cell detection

To detect and quantify neurons labeled with tdTomato from 3D brain images acquired with light-sheet microscopy, we used the cell detection method described in [66]. First, the raw TIFF image was converted into HDF5 format to allow fast and parallel access to the 3D array data on disk. Then, a supervised machine learning method implemented in ilastik [67] was used to classify voxels into three categories: (1) tdTomato-expressing cells (2) fibers labelled with fluorescent proteins or blood vessels with strong autofluorescence and (3) background. Following this classification, single neurons were segmented and fluorescence intensities were quantified using ecc (https://github.com/DSPsleeporg/ecc).

## 4.7 Brain registration

To register cleared brain images to the atlas space, we followed and implemented the method used in the CUBIC pipeline [29,66,68]. We used Allen Mouse Brain Common Coordinate Framework (ABA-CCF) v3 as a reference brain [25]. We downsampled the autofluorescence channel of the cleared brain to make a reduced image with 50 um a voxel size. We then aligned it with the average brain template image of CCFv3 ($50\mu m$ voxel size) using the symmetric image normalization method (SyN) implemented in ANTs library [69]. Since portions of the olfactory bulb (OLF) and cerebellum (CB) were obstructed in our image, we excluded these regions from our analysis. To do this, we manually made image masks to hide OLF and CB from the cleared tissue image. Accordingly, a modified version of ABA-CCFv3 was generated using a custom Python script where OLF and CB were hidden. After computing the warp field to map the cleared brain to the atlas, we applied the same transformations to the table of the detected cells. We then assigned a unique brain region ID to each cell using the annotation image provided by ABA-CCFv3.

## 4.8 Use of spatial transcriptomics data

A highlight of this work is the use of the spatial transcriptomic dataset [6], which provides a comprehensive spatial and molecular profile of approximately 10 million cells in the mouse brain. This dataset is divided into four subsets (Zhuang-ABA1-4), and we utilize the largest subset, containing 2.4 million cells from one hemisphere (Zhuang-ABA1). Among the total population, about 1.3 million cells are neurons, while the remaining non-neuronal cells are excluded during the matching process, as our experimental approach selectively captures c-fos-expressing neurons. To match neurons in our dataset with Zhuang-ABA1, we first identify the regional information for each neuron. Then, for each specific brain subregion, we extract the corresponding neuronal populations from both datasets. Using their spatial coordinates, we apply the K-nearest neighbor (KNN) algorithm to find the closest matching neuron in Zhuang-ABA1 for each neuron in our dataset. As illustrated in Fig 5, we limit this matching process within the same brain region to avoid inaccuracies that may arise when matching neurons near the regional boundaries. Because data in Zhuang-ABA1 only covers one hemisphere, we have also assumed symmetry between the left and right hemispheres during the matching process. After matching, we assume that each neuron in our dataset shares the same molecular and spatial properties as its matched counterpart in Zhuang-ABA1, enabling a detailed analysis of neuronal activity and organization.

## 4.9 Construction and analysis of active connectivity

Please see the appendix.

## 4.10 Data visualization

All single-neuron visualizations (Figs 2–6) in the paper are done using the Python 3D visualization package PyVista [70]. Biorender and Affinity software were used for schematics in Fig 1.

## 4.11 Additional data

1. S1 Data.csv contains the neuron count of different regions of all brains described in this paper.
2. S2 Data.xlsx contains the neuron count of different regions, with further details on neurotransmitter information after matching with the spatial transcriptomic data.
3. S3 Data.xlsx contains the complete active connectivity matrix of different brains, where each matrix is stored in a separate sheet in the file.
   Raw data associated with this study, including neuronal images and coordinates of segmented neurons of each brain, along with the codes, are deposited at the following repository:
   https://figshare.com/s/5c518ccaedc4dd5baeaa.

## Acknowledgments

We thank Ningyuan Wang for helpful discussions on this work and Nikita Vladimirov (mesoSPIM team, ZMB, University of Zurich) for his valuable support with 3D brain imaging.

This paper is dedicated to Steven A Brown.

## Author contributions

**Conceptualization:** Guanhua Sun, Shoi Shi, Koji L Ode, Steven A Brown, Konstantinos Kompotis, Daniel B. Forger.

**Data curation:** Guanhua Sun, Konstantinos Kompotis.

**Formal analysis:** Guanhua Sun.

**Funding acquisition:** Steven A Brown, Hiroki R Ueda, Daniel B. Forger.

**Investigation:** Guanhua Sun, Tomoyuki Mano, Alex Rosi-Andersen, Erica Pedron, Steven A Brown, Konstantinos Kompotis, Daniel B. Forger.

**Methodology:** Guanhua Sun, Tomoyuki Mano, Steven A Brown, Konstantinos Kompotis.

**Supervision:** Steven A Brown, Konstantinos Kompotis, Daniel B. Forger.

**Validation:** Konstantinos Kompotis.

**Visualization:** Guanhua Sun, Alvin Li, Konstantinos Kompotis.

**Writing – original draft:** Guanhua Sun, Tomoyuki Mano, Konstantinos Kompotis, Daniel B. Forger.

**Writing – review & editing:** Guanhua Sun, Koji L Ode, Konstantinos Kompotis, Daniel B. Forger.

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
