## [Editor Report · Decision Letter 0]

29 May 2025

Dear Danny,

Thank you for submitting your revised manuscript entitled "A framework to determine active neurons and networks within the mouse brain" for consideration as a Research Article by PLOS Biology and I am sorry for the delay in getting back to you with an initial decision on your paper. I had wished to discuss your revision with the Academic Editor, but due to an issue with an overactive spam filter, it took a bit longer than normal for me to get in touch with him/her. After now having quickly discussed your paper with the Academic Editor, I am writing to let you know that we would like to send your submission back to the original reviewers for their input.

Once your full submission is complete, your paper will undergo a series of checks in preparation for peer review. After your manuscript has passed the checks it will be sent out for review. To provide the metadata for your submission, please Login to Editorial Manager (https://www.editorialmanager.com/pbiology) within two working days, i.e. by Jun 02 2025 11:59PM.

Kind regards,

Luke

Lucas Smith, Ph.D.

Senior Editor

PLOS Biology

lsmith@plos.org

---

## [Decision Letter · Decision Letter 1]

2 Sep 2025

Dear Danny,

Thank you again for your patience while we considered your revised manuscript "A framework to determine active neurons and networks within the mouse brain" for consideration as a Research Article at PLOS Biology and I apologize again for the protracted review process for your study. Your revised study has now been evaluated by the PLOS Biology editors, the Academic Editor and by one of the original reviewers (reviewer 2). Note that the original reviewers 1 and 3 were not available to re-review your paper. In their place we ended up contacting a new reviewer (reviewer 4) to help assess the revision.

In light of the reviews and our Academic Editor's assessment of your study, which you will find at the end of this email, we are pleased to offer you the opportunity to address remaining points from the reviewers in a revision that we anticipate should not take you very long. We will then assess your revised manuscript and your response to the reviewers' comments with our Academic Editor aiming to avoid further rounds of peer-review, although we might need to consult with the reviewers, depending on the nature of the revisions.

You will see that both of the reviewers agree that the revision has largely addressed the previous reviewer concerns and that the study has been strengthened in the process and will be of interest to the field. However, reviewer 4 has identified a few limitations which should be more clearly discussed, or possibly addressed with additional analyses (if feasible). The Academic Editor has also assessed the revision and agrees that your response to reviewers is good. S/he did have one additional suggestion for analysis that could strengthen the study further. I am including a paraphrased version of their comments below the reviews, for you to address with new analyses, if feasible, or with textual changes.

Below that, I have included a few data and policy related requests. Please be sure to address those as well.

As a last note, in addition to these revisions, you may need to complete some formatting changes, which you will receive in a follow up email. A member of our team will be in touch with a set of requests shortly. If you do not receive a separate email within a few days, please assume that checks have been completed, and no additional changes are required.

We expect to receive your revised manuscript within 1 month, however if you need more time we would be happy to grant an extension. Please email us (plosbiology@plos.org) if you have any questions or concerns, or would like to request an extension.

**IMPORTANT - SUBMITTING YOUR REVISION**

*Resubmission Checklist*

*Published Peer Review*

*PLOS Data Policy*

*Blot and Gel Data Policy*

Sincerely,

Luke

Lucas Smith, Ph.D.

Senior Editor

PLOS Biology

lsmith@plos.org

REVIEWS:

Reviewer #2: The Authors have significantly remodeled the manuscript, increased the number of experiments and time windows, and refined the methods. Most importantly, however, the authors fully addressed the issues regarding the novelty elements of the study by including a mouse spatial transcriptomic dataset, among other actions. I have also inspected the provided TIFF stacks and CSV tables, and they meet the minimum required standards. I have no further remarks regarding the manuscript.

Reviewer #4: In the study the authors "introduced an experimental-computational pipeline to study the active neurons and networks in the mouse brain". I believe this is a very nice and important tool, and the application used is very interesting.

I have to emphasize that I find it difficult as a reviewer to come into the 2nd round of reviews, as asking for extensive new analyses is not ideal at this stage. The authors have done a great job answering the prior reviewer question. However, I believe some important limitations remain, and are not clearly mentioned in the paper. I would like to ask the authors for a few small edits to point them out clearly and help clarify the manuscript.

1. The method presented, while great for homogeneous areas, will likely mix cells and produce statistical averages across transcriptomically heterogeneous areas. In Section 2.4: the areas analyzed are relatively homogeneous transcriptomically. The authors should either include analysis of areas with heterogeneous cell types (e.g. most cortical L2/3 areas), or clearly state this as a limitation clearly at the end of 2.4. This limitation is briefly mentioned later in the discussion, but I think it should be present when presenting the results.

2. As a result of limitation 1, the conclusion that in the cortex there is relative stability of E/I activation can be simply a lack of capacity of the method to tell apart e/i cells in cortex. I did not see a mention to this limitation.

3. The authors mention: "Although a more recent update to this dataset is available [3], providing layer-specific and cell-type-specific projection information, it is limited to the thalamocortical network and therefore was not used for the whole-brain analyses conducted in this study." There is a newer connectivity model (Koelle et al. 2023) which is cell-type specific and extends the work from Harris et al 2019 (the reference 3) to the whole-brain. I presume the authors constructed their framework before the 2023 model was available. If this is the case, please mention it. If not, do please explain why the older model (Knox et al. 2018) was used.

4. There isn't a clear limitations section or paragraph in the discussion, which would help the clarity.

Minor; reference compilation error on Line 412.

The changes mentioned are small enough to not require the paper to be resent to the reviewer.

PARAPHRASED COMMENTS FROM THE ACADEMIC EDITOR:

The Academic Editor agrees with the reviewers that the revised version of the paper 'generally looks quite good' - but s/he has one last suggestion:

"I appreciate that in the Discussion the authors try to contrast what they call active connectivity (which by their definition is a necessary condition for inter-regional communication) with traditionally established functional connectivity, as defined by time series correlations. Given that in the revised work, they consider activity for more time points, they may perhaps also be able to derive a correlation based functional connectivity measure for the data, in order to compare it directly with active connectivity. However, if it is not practical, this direct comparison could also be left for future work and just be mentioned as an outlook."

EDITORIAL DATA AND POLICY REQUESTS

1) FINANCIAL DISCLOSURES: Please update the financial disclosures statement in our editorial manager system to include all of the following details:

-Initials of the authors who received each award

-Grant numbers awarded to each author

-The full name of each funder

-URL of each funder website

-Did the sponsors or funders play any role in the study design, data collection and analysis, decision to publish, or preparation of the manuscript?

2) ETHICS STATEMENT: Please update the ethics statement in your methods section to include the approval number for the animal care and use protocol approved by the Canton of Zurich.

3) DATA: Thank you for providing the data and analyses on figshare.

You may be aware of the PLOS Data Policy, which requires that all data be made available without restriction: http://journals.plos.org/plosbiology/s/data-availability. Note that we do not require all raw data. Rather, we ask that all individual quantitative observations that underlie the data summarized in the figures and results of your paper be made available in one of the following forms:

a. Supplementary files (e.g., excel). Please ensure that all data files are uploaded as 'Supporting Information' and are invariably referred to (in the manuscript, figure legends, and the Description field when uploading your files) using the following format verbatim: S1 Data, S2 Data, etc. Multiple panels of a single or even several figures can be included as multiple sheets in one excel file that is saved using exactly the following convention: S1_Data.xlsx (using an underscore).

b. Deposition in a publicly available repository. Please also provide the accession code or a reviewer link so that we may view your data before publication.

>>Regardless of the method selected, please ensure that you provide the individual numerical values that underlie the summary data displayed in the following figure panels as they are essential for readers to assess your analysis and to reproduce it:

Fig 2B-C; Fig 3A,B; Fig 5A,C-E; Fig 6 A-C; Fig 8F; Fig 9B; Fig 10B-F;

Fig S2D-E; Fig S3A-E; FIig S4A-H; Fig S5C-D; Fig 7 A-G; Fig 8A-f; Fig S10A-J; Fig S11

Sorry if this data is already contained within your figshare deposition and I somehow missed it. please be sure that figure legends in your manuscript include information on where the underlying data can be found, and ensure your supplemental data file/s has a legend. Please ensure that your Data Statement in the submission system accurately describes where your data can be found.

4) CODE: I see your data availability statement says the code will be made available after acceptance. Per journal policy, please do go ahead and provide us with a link to your custom code generated during the course of this investigation. The deposition can be set to private, for now, but will need to be available without restrictions at publication. Please ensure that the code is sufficiently well documented and reusable, and that your Data Statement in the Editorial Manager submission system accurately describes where your code can be found. Please note that we cannot accept sole deposition of code in GitHub, as this could be changed after publication. However, you can archive this version of your publicly available GitHub code to Zenodo. Once you do this, it will generate a DOI number, which you will need to provide in the Data Accessibility Statement (you are welcome to also provide the GitHub access information). See the process for doing this here: https://docs.github.com/en/repositories/archiving-a-github-repository/referencing-and-citing-content

---

## [Editor Report · Decision Letter 2]

17 Oct 2025

Dear Danny,

Thank you for the submission of your revised Research Article "A framework to determine active neurons and networks within the mouse brain" for publication in PLOS Biology and thank you also for addressing the last reviewer and editorial requests in this revision. On behalf of my colleagues and the Academic Editor, Claus C. Hilgetag, I am pleased to say that we can in principle accept your manuscript for publication, provided you address any remaining formatting and reporting issues. These will be detailed in an email you should receive within 2-3 business days from our colleagues in the journal operations team; no action is required from you until then. Please note that we will not be able to formally accept your manuscript and schedule it for publication until you have completed any requested changes.

**IMPORTANT: As you address any production related requests, to come, please also address the following editorial requests:

1 - TITLE: After some discussion within the team, we would suggest a tweak to your title to capture a bit more of what was shown in your paper. If you agree, we suggest you change the title to:

""A framework to determine active neurons and networks within the mouse brain reveals how brain activity changes over the course of the day."

2 - DATA - thank you for providing the underlying data for your paper on figshare. Can you please add a sentence to each relevant figure legend, pointing readers to this file? For example, you could add the sentence "The data underlying this figure can be found at https://figshare.com/s/5c518ccaedc4dd5baeaa"

PRESS

Sincerely, 

Luke

Lucas Smith, Ph.D.

Senior Editor

PLOS Biology

lsmith@plos.org